# Liquid-Crystal and Fire-Retardant Properties of New Hexasubstituted Cyclotriphosphazene Compounds with Two Schiff Base Linking Units

**DOI:** 10.3390/molecules25092122

**Published:** 2020-05-01

**Authors:** Zuhair Jamain, Melati Khairuddean, Tay Guan-Seng

**Affiliations:** 1Faculty of Science and Natural Resources, Universiti Malaysia Sabah (UMS), Kota Kinabalu 88400, Malaysia; 2School of Chemical Sciences, Universiti Sains Malaysia (USM), Penang 11800, Malaysia; 3School of Industrial Technology, Universiti Sains Malaysia (USM), Penang 11800, Malaysia; taygs@usm.my

**Keywords:** schiff base, cyclotriphosphazene, liquid-crystal, fire-retardant, non-mesogenic

## Abstract

A series of new hexasubstituted cyclotriphosphazene compounds (**4a**–**j**) consisting of two Schiff base linking units and different terminal substituents was successfully synthesized and characterized. The structures of these compounds were confirmed using Fourier Transform Infra-Red (FTIR), Nuclear Magnetic Resonance (NMR), and CHN elemental analysis. Polarized optical microscopy (POM) was used to determine their liquid-crystal behavior, which was then further confirmed using differential scanning calorimetry (DSC). Compounds **4a**–**i** with heptyl, nonyl, decyl, dodecyl, tetradecyl, hydroxy, 4-carboxyphenyl, chloro, and nitro terminal ends, respectively, showed the liquid-crystal properties, whereas compound **4j** with the amino group was found to be non-mesogenic. The attachment of an electron-donating group in **4j** eventually give a non-mesogenic product. The study of the fire-retardant properties of these compounds was done using the limiting oxygen index (LOI). In this study, polyester resin (PE) was used as a matrix for moulding, and the LOI value of pure PE was 22.53%. The LOI value increased to 24.71% when PE was incorporated with 1 wt.% of hexachlorocyclotriphosphazene (HCCP), thus indicating that HCCP has a good fire-retardant properties. The result showed that all the compounds have good agreement in their LOI values. Compound **4i** with a nitro terminal group gave the highest LOI value of 28.37%.

## 1. Introduction

Hexachlorocyclotriphosphazene (HCCP) is an inorganic phosphorus–nitrogen compound consisting of alternating phosphorus and nitrogen atoms, bound to each other through alternating single and double bonds [1]. HCCP derivatives are excellent models for structure–activity studies and their multiarmed rigid ring allows the exploration of new discotic molecules in the field of liquid crystals [2,3]. Liquid crystal is an intermediate phase between liquid and solid, a state of matter that has both the properties of isotropic liquid and of solid crystal. As temperature increases, the solid absorbs some heat and melts into a liquid-crystal phase whereby molecules can move freely, disrupting the positional order without changing the orientational order [4].

To study the behavior of the HCCP compound, which consists of inorganic backbones as well as organic side chains, small changes in the structure must be made [5,6]. Due to the high reactivity of the P–Cl bond, the corresponding substitution method allows the introduction of a wide range of substituents [7,8] and hence provides substituted cyclotriphosphazene derivatives with different chemical and physical properties [9,10]. Such compounds have been used in commercial applications [11,12,13,14] especially in liquid-crystal and fire-retardant materials, where it has been proved that phosphazene-based compounds exhibit effective flame-retardant properties [15]. Jae Shin et al. (2010) reported the application of cyclotriphosphazene derivatives as fire retardants [16]. Phosphorus-based materials are thermally quite stable and versatile in their fire-retardant activity. Interestingly, compounds with a phosphorus atom often exhibit both condensed and gas phase state [17]. The chemical reactions that take place in the condensed phase at elevated temperature, such as hydrolysis, dehydration, and chain scission, produce carbonaceous char residues on the surface of the phosphorus-based materials, which retards the material from further burning [18,19].

Even though extensive research has focused on the use of substituted cyclotriphosphazene derivatives on fire retardants, the study of cyclophosphazene-based compounds on liquid crystal has still not being fully explored [20,21]. The HCCP molecule is a well-known compound with good thermal stability and fire retardancy, which is due to the hexa-functionality and high phosphorus content. Modification of the HCCP core system with organic side arms allows the exploration of new discotic molecules in the field of liquid crystals with fire-retardant properties. The incorporation of HCCP into organic compounds with a Schiff base linking unit is done to increase the resistance of the material towards ignition. The different terminal substituents in the side arms will gain a better insight of the structure–properties relationship of these types of compounds. The research into liquid-crystal materials with fire-retardant properties with flexible ordered structures has led to the discovery of new properties and applications such as advanced technological devices or composite applications. Extensive exploration of these materials can detect small changes in temperature, electromagnetic radiation, mechanical stress, and chemical environment [22,23,24].

This research focused on the preparation of a series of hexasubstituted cyclotriphosphazene derivatives with two Schiff base linking units. Schiff base contains carbon double bonds with nitrogen atoms connected to an alkyl or aryl group but not the hydrogen. These Schiff base molecules provide a stepped core structure that can maintain the molecular linearity. This linearity will provide high stability and enable mesophase formation [25]. Moreover, Schiff base linking units were found to enhance the fire-retardant properties of polyester resin due to their thermal stability [26]. During the burning process, this molecule can transform into a cross-linked structure, which promotes the formation of char on the surface in the condensed phase. These char layers shield the surface of the polyester resin from combustion [27]. Therefore, both liquid-crystal and fire-retardant properties of the cyclotriphosphazene core system attached to a Schiff base linking unit were investigated in this study.

In this work, polyester resin (PE) was used as a matrix for moulding in order to study the fire-retardant properties of these compounds. PE is an unsaturated synthetic resin and can be considered to be a combustible material [28]. This resin produces a lot of heat during combustion, due to low thermal stability, and becomes a potential fire risk. Thus, the modification of polyester resin by mixing with cyclotriphosphazene-based compounds with fire-retardant properties is necessary to overcome this problem.

## 2. Results and Discussion

### 2.1. Synthesis of the Intermediates and Final Compounds

Intermediates **1a**–**e** were synthesized using the alkylation reaction of 4-hydroxybenzaldehyde with different alkyl bromides (heptyl, nonyl, decyl, dodecyl, and tetradecyl). Intermediates **2a**–**i** were formed from the condensation reaction of **1a**–**e** and other commercially para-substituted benzaldehydes (-OH, -COOH, -Cl, and -NO_2_), respectively. Reaction of hexachlorocyclotriphosphazene, HCCP, with 4-hydroxybenzaldehyde gave intermediate **3**, which was then used for further reaction to yield the final compounds **4a**–**i**. Finally, the reduction of compound **4i** afforded compound **4j**. The overall synthesis pathway for hexasubstituted cyclotriphosphazene compounds with different terminal ends in the side arms bearing two Schiff base linking units is shown in Scheme 1, Scheme 2 and Scheme 3.

All the synthesized intermediates (**1a**–**e**, **2a**–**i**, and **3**) and final compounds **4a**–**j** were then characterized using Fourier Transform Infra-Red (FTIR), Nuclear Magnetic Resonance (NMR), and CHN elemental analysis. Liquid-crystal properties of these compounds were determined using polarized optical microscope (POM) and further confirmed using differential scanning calorimetry (DSC). In this study, the limiting oxygen index (LOI) test is used to determine the minimum amount of oxygen required to support the combustion of a sample. The compact data of these intermediates and compounds are summarized in Section 3.3.

### 2.2. FTIR Spectral Data of the Intermediates and Final Compounds

The IR data for intermediates **1a**–**e** with heptyl, nonyl decyl, dodecyl, and tetradecyl chains, respectively, showed a similar pattern to the absorption bands at 2850 and 2930 cm^−1^ for the symmetrical and asymmetrical Csp^3^–H stretching, 2733 cm^−1^ for the C–H stretching of the aldehyde, 1690 cm^−1^ for C=O stretching, 1600 cm^−1^ for C=C aromatic stretching, and 1250 cm^−1^ for C–O stretching. The absence of the O–H stretching at 3300 cm^−1^ indicated that the insertion of the alkyl groups was a success.

These intermediates **1a**–**e** and other commercially available aldehydes such as 4-hydroxybenzaldehyde, 4-formylbenzoic acid, 4-chlorobenzaldehyde, and 4-nitrobenzaldehyde were used in the reaction with 1,4-phenylenediamine to form the Schiff base intermediates, **2a**–**i**. The IR data of intermediates **2a**–**i** showed similar absorption bands with two bands at 3320 and 3450 cm^−1^ for the N–H stretching. Other bands at 1610, 1550, and 1165 cm^−1^ were assigned to the C=N, C=C, and C–N stretching, respectively. The absorption band for C–O stretching at 1250 cm^−1^ can also be observed for intermediates **2a**–**g**. Only intermediates **2a**–**e** displayed the absorption bands of asymmetrical and asymmetrical Csp^3^-H stretching at 2850 and 2920 cm^−1^, while intermediates **2f** and **2g** showed a band at 3210 cm^−1^ for O–H stretching. An additional band at 1698 cm^−1^ for C=O stretching was observed for **2g** and band at 790 cm^−1^ for C–Cl stretching for **2h**. The band for aldehydic C–H stretching was not observed in any of the intermediates, **2a**–**i**, which confirmed the success of the condensation reaction.

The substitution reaction of HCCP with *p*-hydroxybenzaldehyde formed intermediate **3.** The main absorption bands for **3** were observed at 2730 (H-C=O), 1700 (C=O), 1595 (C=C), and 1205 cm^−1^ (C–O). The absorption bands from the cyclotriphosphazene ring, P=N stretching, appeared at 1151 cm^−1^ while P–O–C bending was located at 944 cm^−1^. Intermediate **3** was reacted with **2a**–**i** to afford final compounds with two Schiff base linking units, **4a**–**i**.

According to the IR spectra of compounds **4a**–**j** (Figure 1), the presence of the alkoxy moieties in compounds **4a**–**e** was substantiated by the appearance of the absorption bands at 2851 and 2920 cm^−1^ (Csp^3^–H stretching). The absorption bands at 1168, 1248, and 1509 cm^−1^ were assigned to the stretching of C–N, C–O, and aromatic C=C, respectively. The stretching of P=N and P–O–C bending of the cyclotriphosphazene ring in the compound was observed at 1192 and 983 cm^−1^, respectively. Only compound **4g** showed the absorption band for carbonyl (C=O) of the 4-carboxyphenyl substituent at 1673 cm^−1^, while C–Cl bending at 823 cm^−1^ was observed in compound **4h**. On the other hand, the reduction reaction of compound **4i** to form **4j** was completed with the appearance of two absorption bands at 3312 and 3478 cm^−1^ for the N–H stretching in the amino group. The stretching corresponding to C=N linkage at 1616 cm^−1^ in the IR spectra of compounds **4a**–**j** indicated that the Schiff base formation was successful.

### 2.3. NMR Spectral Data of Final Compounds

Compound **4a** was used to represent the structure confirmation in the series. The structure of compound **4a** with complete atomic numbering is shown in Figure 2.

^1^H NMR spectrum of compound **4a** (Figure 3) showed two different azomethine proton signals, which were observed in the downfield region at δ 8.66 (H5) and δ 8.55 ppm (H10), respectively. Six doublets integrating to 12 aromatic protons appeared in the range of δ 7.03–7.95 ppm and were assigned to the para-substituted aromatic protons. The presence of the azomethine groups caused the signals of H3 and H12 to be the most deshielded aromatic protons. Meanwhile, the signals for H7 and H8 almost overlapped as the protons experienced similar chemical environment. Moreover, proton signals of the aliphatic chains were observed in the upfield region, with the oxymethylene protons (H15) more deshielded among other methylene protons due to their close proximity to the neighboring electronegative oxygen atom. Therefore, the signal of these protons appeared at δ 4.07 ppm as a triplet while that of the other methylene protons (H16–H20) were observed in the region of δ 1.30–1.79 ppm. A triplet in the most upfield (δ 0.89 ppm) region was assigned to H21.

The ^13^C NMR spectrum of compound **4a** as shown in Figure 4a indicates that **4a** has 21 carbon signals in the side arms. These signals consist of two azomethine, six aromatics, six quaternary, six methylene, and one methyl carbons. Further complete assignment of all carbons was done using DEPT experiments. DEPT 90 gave information on methine carbon (CH) and DEPT 135 gave information on the methylene carbon (CH_2_) which appeared as the negative signal while methine (CH) and methyl (CH_3_) carbon showed positive signals.

Based on the DEPT 90 spectrum (Figure 4b), peaks at δ 159.38 and 158.83 ppm were assigned to two azomethine carbons, C5 and C10. The aromatic carbon signals were observed at δ 115.46, 122.13, 122.32, 129.28, 130.53, and 130.81 ppm which can be assigned to C12, C8, C7, C3, C13, and C2, respectively. Six carbon signals, which were absent in the DEPT 90 spectrum were assigned for the quaternary carbons (C1, C4, C6, C9, C11, and C14). C1 and C14 were in the most deshielded region at δ 162.09 and 162.02 ppm, respectively, due to electronegativity effect of the oxygen atom. On the other hand, C4 and C11 were assigned at δ 136.49 and 135.76 ppm, respectively, as these carbons were attached to the less electronegative atoms among other quaternary carbons. The two carbons attached to nitrogen atoms, C6 and C9, were assigned to peaks at δ 150.63 and 149.14 ppm, respectively. Six methylene carbons of the heptyl chains (C15–C20) showed negative signals in the DEPT 135 spectrum (Figure 4c). C15 adjacent to the oxygen atom was observed at δ 68.64 ppm. Five methylene carbons of the aliphatic chains showed signals at δ 22.31 (C20), 25.88 (C19), 28.76 (C18), 29.16 (C17), and 31.58 ppm (C15). The signal at δ 13.91 ppm disappeared in the DEPT 90 spectrum but appeared in the positive region of DEPT 135 spectrum, confirming that this carbon signal belongs to the methyl carbon (C21).

The assignment of all proton (^1^H–^1^H) correlations between proton and neighboring proton was further confirmed using a COSY (^1^H–^1^H) NMR experiment. According to the COSY (^1^H–^1^H) spectrum (Figure 5), H5 and H6 did not show any correlation with other protons, which indicated that there are the azomethine protons. The correlation of the aromatic protons can be seen between H2 and H3, H7 and H8, and H12 and H13, respectively. However, the correlations between methylene protons of the heptyl chains could be observed between H15 and H16, H16, and H17, and H18–H21.

Assignment of the protonated carbons in compound **4a** was done using HSQC NMR experiment (Figure 6). The azomethine protons at δ 8.66 (H5) and δ 8.55 ppm (H10), showed correlations with the carbons at δ 159.38 (C5) and 158.83 (C10) ppm, respectively. The ^1^H–^13^C connectivity in the aromatic region could be observed between H2 and C2 at δ 130.81 ppm; H3 and C3 at δ 129.28 ppm; H7 and C7 at δ 122.32 ppm; H8 and C8 at δ 122.13 ppm; H12 and C12 at δ 115.56 ppm; and H13 with C13 at δ 130.53 ppm. Moreover, H15 at δ 4.07 ppm was correlated with C15 at δ 68.64 ppm. The connectivity of methylene protons (H16–H20) with their corresponding carbons (C16–C20) was also observed in the HSQC spectrum. A triplet at δ 0.89 ppm assigned to H21 for the methyl protons showed connectivity with its methyl carbon (C21) at δ 14.06 ppm in the most upfield region. All the data obtained from the COSY (^1^H–^1^H) and HSQC (^1^H–^13^C) spectra are summarized in Table 1.

The ^31^P NMR spectrum of compound **4a** (Figure 7a) showed only a singlet at δ 8.20 ppm. The peak of this compound was shifted to the upfield region as compared to that of **HCCP** with six electron-withdrawing chlorine atoms (Figure 7b), indicating that all the phosphorus had been substituted with the same side arms. The ^31^P NMR spectrum showed the chemical shifts of the hexasubstituted cyclotriphosphazene series experience more shielding. This behavior was attributed to the molecular structure of the hexa-series that contains six side arms with high electron density. As a result, greater shielding effect was observed.

Furthermore, other homologues with alkylated terminal chains (compounds **4b**–**e**) showed similar splitting patterns and chemical shifts in the ^1^H and ^13^C NMR spectra with compound **4a** but only differed in terms of the number of protons and carbons in the alkyl chains. Compounds **4f**–**j** with small terminal substituents (OH, Cl, COOH, NO_2_, and NH_2_) showed two singlets of azomethine and six doublets for aromatic protons in the ^1^H NMR spectra. There was a slight difference in the chemical shift values of the signals of each compound due to the chemical environment, electronegativity effect, and bond angles. Only compounds **4f** and **4g** showed the hydroxyl and carboxyl proton peaks in the ^1^H NMR spectra.

### 2.4. Determination of Liquid-Crystal Properties Using POM

The phase textures of intermediate and final compounds were determined using POM. In this study, all the intermediates (**1a**–**e**, **2a**–**j**, and **3**) were found to be non-mesogenic with no liquid-crystal behavior. Meanwhile, compounds **4a**–**i** with two Schiff base linking units were found to exhibit liquid-crystal phase while compound **4j** was found to be non-mesogenic. Observation under POM (Figure 8) showed that compound **4a** exhibited the thread-like nematic phase with four-point brushes in the cooling cycle. Further cooling changed the nematic phase into SmA phase before it became a crystal phase.

The phase transitions of compounds **4b**–**d** showed the focal conic fan texture of SmA phase in the heating and cooling cycles, as illustrated in Figure 9. Meanwhile, compound **4e** exhibited two different types of phase transition in both heating and cooling cycles. Upon cooling, a focal conic fan of SmA phase was formed from the isotropic phase and further cooling resulted in the formation of a broken focal conic fan of SmC phase which might be due to the stacking of molecules. The texture is illustrated in Figure 10.

For compounds **4f**-**i**, schlieren texture of the nematic phase with four-point brushes was clearly observed under POM (Figure 11 and Figure 12) in the heating and cooling cycles. Upon cooling, compound **4f** showed the formation of droplets from the isotropic phase which became the nematic phase with a thread-like texture, as shown in Figure 11. However, compound **4j** without any liquid-crystal properties only displays the phase transition of crystal to isotropic phase in heating cycle and isotropic to crystal phase in cooling cycle.

### 2.5. Determination of Thermal Transitions Using DSC

Texture observation under POM with controlled temperature helped to determine the type of mesophases present in the molecules. Only compounds exhibiting mesophase(s) were sent for DSC measurements. The DSC thermograms are used to confirm the phase transition temperatures and enthalpy change involved during the transition in the heating and cooling cycles. The phase transitions and its corresponding enthalpy change of compounds **4a**–**i** with two Schiff base linking units is summarized in Table 2.

The DSC thermogram of compounds **4a** and **4e** shows three endotherms or curves with their enthalpy values, in both heating and cooling cycles. Upon heating of compound **4a**, the curves were attributed to the phase transitions from crystal to SmA and nematic before reaching a clearing temperature at 235.89 °C. Similar phase transitions were also observed in the reversed order of the cooling cycle. For compound **4e**, the phase transitions were from the crystal to SmC and then SmA before it became the isotropic phase with the melting point observed at 147.75 °C and the clearing temperature at 197.81 °C.

Compounds **4b**–**d** and **4f**–**i** exhibited only two curves in the DSC thermogram which indicates that these compounds have only one liquid-crystal phase. The DSC thermogram of compounds **4b**–**d** displayed the transition from crystal to SmA and isotropic phase in the heating cycle. Meanwhile, compounds **4f**–**i** with substituents such as hydroxy, 4-carboxyphenyl, chloro, and nitro terminal chains in the side arms also showed two curves in the DSC thermogram, the transition from crystal to the thread-like nematic, and isotropic phase. In the cooling cycle, these two curves of phase transitions were also observed in the reverse order. The DSC thermogram of compound **4f** is shown in Figure 13. In addition, the DSC thermogram of compounds **4a**–**e** and **4g**–**i** are provided in the Appendix A Section. The existence of endotherm appears in the DSC thermogram for each of the mesophase transition, which corresponds to the texture of liquid crystal, which is observed under POM.

### 2.6. Structure–Properties Relationship

The study on the relationship between structure and liquid-crystal mesophase behavior is very important in designing new liquid-crystal materials with desirable properties for future applications.

Based on the POM observation, all the final compounds in the series with a Schiff base linking unit that bore a different terminal alkoxy function were mesogenic and the wider of the thermal mesomorphic range increased with increasing chain length. Compound **4a** with heptyl chains showed smectic and nematic phases with the texture of thread-like four-point brushes. As the length of the alkyl chain increased, the tendency for formation of the nematic phase was reduced. This trend was observed for compounds **4b**–**e** which exhibited only the smectic phase since the flexibility of the long chains tends to decrease the clearing temperatures, T_c_. Smectic phases are more ordered than the nematic phase. Nematic phase has the positional order but no orientational order and closest to the isotropic phase. Hence, effective molecular packing is essential for the formation of stable phases. Moreover, the polarisability of the molecules also increased with an increase of the alkoxy chain length in the compounds. This enhanced the cohesive forces between the sides and the core of the molecules, thus increased the tendency to form the smectic layer [35]. Molecules with smectic phase behavior are likely to favor a lamellar packing, which is due to higher Van der Waals interactions and intertwining possibility between the alkoxy chains. Thus, the nematic orientation cannot be adopted as the chain length increased and resulted in only a smectic phase in compounds **4b**–**e**.

In this work, compounds **4a**–**e** exhibited the smectic A phase while compound **4e** showed an additional of smectic C. The presence of smectic C phase in compound **4e** is due to the tilted analogue of the smectic A phase [36]. In smectic C, molecules are aligned with their long axes tilted relative to the layer normal in which the molecules are stacked. Meanwhile, molecules in smectic A are arranged in layers, with long axes perpendicular to the layers which can slide over one another [37].

Small terminal substituents, such as hydroxy or carboxy, do not always induce liquid-crystal behavior [38]. In contrast, Sudhakar et al. (2000) reported that small molecules often exhibit one or more liquid crystalline phases since these molecules have geometrical anisotropy and high polarisability [39]. In this study, compound **4f** with the –OH terminal group and compound **4g** with –COOH substituent showed the formation of the nematic phase. Meanwhile, compound **4h** with chlorine at the terminal end also exhibited the nematic phase. The result agreed with the literature data which demonstrated that compounds with the polar group are more likely to exhibit nematic phase compared to compounds with a methyl group or hydrogen atom at the terminal chain [40]. Chlorine is a polar substituent which possesses a strong dipole moment, thus enhances the stability of the lattice and melting temperatures [41].

The effect of the NO_2_ and NH_2_ groups at the terminal end in promoting the mesophase properties was also highlighted. The nematic phase exhibited by compound **4i** was due to the high polarity of NO_2_ group which increases the molecular aromaticity and polarisability of the molecules [42]. On the other hand, compound **4j** was found to be non-mesogenic. This phenomenon might be due to the properties of the NH_2_ group as an electron-donating group, which increases the repulsive interactions between adjacent aromatic rings. As a result, the mesophase transitions cannot be induced. Galewski and Coles (1999) also reported that the attachment of an electron withdrawing at the terminal end of a compound induces the liquid-crystal properties, while compounds with electron-donating groups did not show any liquid-crystal properties [43].

In this research, the effect of the Schiff base (–C=N–) linking unit was also studied. When the Schiff base linking unit conjugated with the phenylene rings, the length of the molecules increased which enhanced the anisotropic polarisability and the flexibility of the compounds [44,45]. Thus, the rigidity and linearity of its constituents were maintained and impart various property alterations to mesomorphic materials.

The Schiff base linkage provides a stepped core structure which can maintain the linearity of the molecules to provide high stability [46]. This induces mesophase formation whereby the phase transition temperatures and the physical properties are usually governed by the linking group [47]. This was proven when all the compounds, **4a**–**i** in this series (two Schiff base linking units) except for compound **4j** with the terminal NH_2_ substituent, showed liquid-crystal behavior. Schiff base offers the possibility of controlling the alignment and orientation of their molecules which can generate liquid-crystal materials [48]. The POM observation of compounds **4a**–**j** is summarized in Table 3.

### 2.7. Determination of Fire-Retardant Properties Using LOI Testing

All the hexasubstituted cyclotriphosphaze compounds **4a**–**j** were further tested for their fire-retardant properties. All the samples were prepared using 1 wt.% to achieve the highest fire retardancy with less additive usage. Their effect is to reduce the initiation of a fire by delaying the spread of flame and provide resistance to ignition. LOI was used to determine the fire-retardant properties of the sample. This test was conducted by suspending the sample vertically inside a closed chamber where this chamber was equipped with oxygen and nitrogen gas inlets so that the atmosphere in the chamber can be controlled. The sample was then ignited from the top and the atmosphere was adjusted to determine the minimum amount of oxygen required to burn a sample at a certain time.

In this study, polyester resin has been used as a matrix for moulding. As shown in Table 4, the LOI value of pure polyester resin was determined as 22.53%. When the polyester resin was incorporated with 1 wt.% of HCCP in the matrix, the LOI value increased to 24.71%. HCCP material is well known as the compound that has high thermal stability and fire retardancy. This behavior was due to its hexa-functionality and high phosphorus content [49].

All compounds showed good agreement in the LOI results where they could be considered to be fire-retardant materials. Generally, materials with LOI value above 26% are considered to have fire-retardant property and will show self-extinguishing behavior [50,51]. The best result was achieved for compound **4i** with nitro (-NO_2_) and compound **4h** with chlorine (-Cl) groups at the terminal end, with LOI values of 28.37% and 27.90%, respectively. This might be due to the electron-withdrawing properties of both the nitro and chlorine groups which induced the properties of fire retardancy. These groups can release the electron from their resonance effects due to their corresponding P–N bonds. Thus, the P–N synergistic effect was enhanced, and they exhibited both the condensed and gas phase action [52]. Similar trends can be observed for the compounds attached to hydroxy, carboxy, and amino groups at the terminal end. However, the LOI values decreased as the alkyl chain length increased. Moreover, the LOI values of the alkylated compounds were lower compared to those of compounds with the hydroxy, carboxy, and amino groups at the terminal end. The longer the alkyl tails are attached to the cyclotriphosphazene ring, the more combustible the target materials [53,54].

## 3. Materials and Methods

### 3.1. Chemicals

The chemicals and solvents used in this study are 1-bromoheptane, 1-bromononane, 1-bromodecane, 1-bromododecane, 1-bromotetradecane, 4-hydroxybenzaldehyde, 4-chlorobenzaldehyde, 4-nitrobenzaldehyde, 4-formylbenzoic acid, 1,4-phenylenediamine, phosphonitrilic chloride trimer, sodium sulphide hydrate, potassium carbonate, potassium iodide, anhydrous sodium sulphate, glacial acetic acid, methanol, ethanol, acetone, *N,N*-dimethylformamide, dichloromethane, *n*-hexane, ethyl acetate, and triethylamine. All the chemicals and solvents were used as received without further purification and purchased from Merck (Darmstadt, Germany), Qrëc (Asia) (Selangor, Malaysia), Sigma–Aldrich (Steinheim, Germany), Acros Organics (Geel, Belgium), Fluka (Shanghai, China), and BDH (British Drug Houses) (Nichiryo, Japan).

### 3.2. Instruments

In this research, Thin Layer Chromatography (TLC) is used to monitor a reaction progress and identify a product in a mixture. The ratios used were 5:95, 10:90, 15:85, 20:80, and 25:75 of ethyl acetate:hexane. Meanwhile, all the synthesized intermediates and compounds were characterized using FTIR spectroscopy (PerkinElmer, Waltham, MA, USA), NMR spectroscopy (Bruker, Coventry, UK), and CHN elemental analysis (PerkinElmer, Waltham, MA, USA). Moreover, the mesophase texture of these compounds was determined using POM (Linkam, London, UK) and their transition was further confirmed using DSC (PerkinElmer, Waltham, MA, USA). In addition, LOI (S.S. Instruments Pvt. Ltd., Delhi, India) was used to determine the minimum amount of oxygen needed to support the combustion of a sample. The sample was prepared by mixing 1 wt.% of the final compound with polyester resin. About 1 wt.% of methyl ethyl ketone peroxide (MEKP) curing agent was added to the mixture and stirred until the sample is homogeneous and then poured into the moulds. The samples were cured for 5 h in an oven at 60 °C and left overnight at room temperature before it was burned using LOI testing. The LOI test was performed using an FTT oxygen index, according to BS 2782: Part 1: Method 141 and ISO 4589 with the dimension of 120 mm × 10 mm × 4 mm. The minimum value of oxygen obtained was expressed as a percentage and the LOI results were calculated according to the equation given below:
LOI = *C_F_* + (k × d)(1)
where: *C_F_* is the oxygen concentration of the final test, k is the factor obtained from the manual book Fire Testing Technology (ISO 4589), and d is the oxygen concentration increment.

### 3.3. Syntheses

#### 3.3.1. Synthesis of 4-Alkoxybenzaldehyde, **1a**–**e**

4-Heptyloxybenzaldehyde, **1a**: 4-Hydroxybenzaldehyde (12.21 g, 0.10 mol) was dissolved in *N,N*-dimethylformamide, DMF (20 mL) while 1-bromoheptane (17.89 g, 0.10 mol) was dissolved in DMF (20 mL), separately. Both solutions were mixed in a 250 mL round bottom flask. Potassium carbonate (20.73 g, 0.15 mol) and potassium iodide (1.66 g, 0.01 mol) were added to the mixture and was then refluxed for 12 h. The reaction progress was monitored using TLC. Upon completion, the mixture was poured into 500 mL cold water and was extracted using dichloromethane (3 × 30 mL). The organic layers were collected, dried with anhydrous sodium sulphate, filtered, and evaporated overnight to form a yellowish oil. The same method was used to synthesis **1b**–**e**. Yield: 18.71 g (85.05%), light-yellow oil. FTIR (cm^−1^): 2929 and 2857 (C*sp^3^*-H stretching), 2738 (H-C=O aldehydic stretching), 1687 (C=O stretching), 1601 (C=C stretching), 1254 (C-O stretching). ^1^H-NMR (500 MHz, CDCl_3_) δ, ppm: 9.81 (s, 1H), 7.76 (d, *J* = 10.0 Hz, 2H), 6.93 (d, *J* = 10.0 Hz, 2H), 3.97 (t, *J* = 5.0 Hz, 2H), 1.72–1.78 (m, 2H), 1.37–1.43 (m, 2H), 1.23–1.34 (m, 6H), 0.84 (t, *J* = 7.5 Hz, 3H). ^13^C-NMR (125 MHz, CDCl_3_) δ, ppm: 190.61, 164.24, 131.88, 129.76, 114.71, 68.38, 31.71, 29.03, 28.96, 25.89, 22.54, 14.01.

4-Nonyloxybenzaldehyde, **1b**: Yield: 20.11 g (81.09%), light-yellow oil. FTIR (cm^−1^): 2924 and 2854 (C*sp^3^*-H stretching), 2733 (H-C=O aldehydic stretching), 1689 (C=O stretching), 1601 (C=C stretching), 1254 (C-O stretching). ^1^H-NMR (500 MHz, CDCl_3_) δ, ppm: 9.79 (s, 1H), 7.73 (d, *J* = 5.0 Hz, 2H), 6.90 (d, *J* = 5.0 Hz, 2H), 3.95 (t, *J* = 7.50 Hz, 2H), 1.70–1.76 (m, 2H), 1.36–1.42 (m, 2H), 1.20–1.32 (m, 10H), 0.82 (t, *J* = 5.0 Hz, 3H). ^13^C-NMR (125 MHz, CDCl_3_) δ, ppm: 190.56, 164.24, 131.87, 129.77, 114.71, 68.37, 31.83, 29.47, 29.32, 29.21, 29.04, 25.93, 22.62, 14.04.

4-Decyloxybenzaldehyde, **1c**: Yield: 21.32 g (81.37%), light-yellow oil. FTIR (cm^−1^): 2924 and 2854 (C*sp^3^*-H stretching), 2733 (H-C=O aldehydic stretching), 1689 (C=O stretching), 1601 (C=C stretching), 1254 (C-O stretching). ^1^H-NMR (500 MHz, CDCl_3_) δ, ppm: 9.83 (s, 1H), 7.77 (d, *J* = 10.0 Hz, 2H), 6.94 (d, *J* = 10.0 Hz, 2H), 3.99 (t, *J* = 7.50 Hz, 2H), 1.74–1.79 (m, 2H), 1.39–1.45 (m, 2H), 1.21–1.33 (m, 12H), 0.84 (t, *J* = 7.5 Hz, 3H). ^13^C-NMR (125 MHz, CDCl_3_) δ, ppm: 190.65, 164.27, 131.92, 129.78, 114.74, 68.41, 31.87, 29.52, 29.32, 29.29, 29.05, 25.95, 22.65, 14.06.

4-Dodecyloxybenzaldehyde, **1d**: Yield: 23.56 g (83.25%), light-yellow oil. FTIR (cm^−1^): 2921 and 2854 (C*sp^3^*-H stretching), 2733 (H-C=O aldehydic stretching), 1692 (C=O stretching), 1601 (C=C stretching), 1254 (C-O stretching). ^1^H-NMR (500 MHz, CDCl_3_) δ, ppm: 9.82 (s, 1H), 7.77 (d, *J* = 10.0 Hz, 2H), 6.94 (d, *J* = 10.0 Hz, 2H), 3.98 (t, *J* = 7.50 Hz, 2H), 1.74–1.79 (m, 2H), 1.39–1.45 (m, 2H), 1.22–1.33 (m, 16H), 0.84 (t, *J* = 7.5 Hz, 3H). ^13^C-NMR (125 MHz, CDCl_3_) δ, ppm: 190.63, 164.26, 131.91, 129.78, 114.73, 68.41, 31.90, 29.64, 29.62, 29.57, 29.53, 29.33, 29.05, 25.95, 22.67, 14.07.

4-Tetradecyloxybenzaldehyde, **1e**: Yield: 24.75 g (79.58%), mp: 56.6–58.7 °C, white powder. FTIR (cm^−1^): 2916 and 2845 (C*sp^3^*-H stretching), 2733 (H-C=O aldehydic stretching), 1689 (C=O stretching), 1601 (C=C stretching), 1251 (C-O stretching). ^1^H-NMR (500 MHz, CDCl_3_) δ, ppm: 9.85 (s, 1H), 7.80 (d, *J* = 10.0 Hz, 2H), 6.96 (d, *J* = 10.0 Hz, 2H), 4.01 (t, *J* = 7.50 Hz, 2H), 1.76–1.81 (m, 2H), 1.41–1.47 (m, 2H), 1.23–1.36 (m, 20H), 0.86 (t, *J* = 7.5 Hz, 3H). ^13^C-NMR (125 MHz, CDCl_3_) δ, ppm: 190.76, 164.29, 131.97, 129.79, 114.76, 68.45, 31.92, 29.68, 29.66 29.64, 29.57, 29.53, 29.34, 29.33, 29.06, 25.95, 22.67, 14.09. CHN elemental analysis: Calculated for C_21_H_34_O_2_: C: 79.19%, H: 10.76%; Found: C: 78.91%, H: 10.66%.

#### 3.3.2. Synthesis of *N*-(4-Substitutedbenxylidene)benzene-1,4-diamine, **2a**–**i**

*N*-(4-heptyloxybenxylidene)benzene-1,4-diamine, **2a**: A mixture of 1,4-phenylenediamine (1.08 g, 0.01 mol) and intermediate **1a** (1.54 g, 0.007 mol) was dissolved in 15 mL methanol in a 100 mL round bottom flask. The reaction was stirred at room temperature for 2 h. The reaction progress was monitored by TLC. Upon completion, the mixture was cooled in ice water and the precipitate formed was filtered and dried. The precipitate was recrystallised from ethanol to yield a yellow powder. The same method was used to synthesis **2b**–**i**. *Yield* = 1.85 g (85.25%), mp: 136.5–138.3 °C, yellow powder. FTIR (cm^–1^): 3462 and 3336 (N-H stretching), 2927 and 2857 (C*sp^3^*-H stretching), 1603 (C=N stretching), 1509 (aromatic C=C stretching), 1248 (C-O stretching), 1168 (C-N stretching). ^1^H-NMR (500 MHz, DMSO-d_6_) δ, ppm: 8.47 (s, 1H), 7.79 (d, *J* = 10.0 Hz, 2*H*), 7.06 (d, *J* = 5.0 Hz, 2H), 7.00 (d, *J* = 10.0 Hz, 2H), 6.64 (d, *J* = 10.0 Hz, 2H), 6.42 (s, 2H), 4.06 (t, *J* = 7.50 Hz, 2H), 1.73–1.78 (m, 2H), 1.43–1.48 (m, 2H), 1.28–1.40 (m, 6H), 0.90 (t, *J* = 7.5 Hz, 3H). ^13^C-NMR (125 MHz, DMSO-d_6_) δ, ppm: 161.32, 154.63, 147.57, 141.36, 139.43, 130.03, 122.37, 115.38, 115.07, 68.58, 31.57, 29.18, 28.74, 25.88, 22.30, 14.07. CHN elemental analysis: Calculated for C_20_H_26_N_2_O: C: 77.38%, H: 8.44%, N: 9.02%; Found: C: 77.05%, H: 8.47%, N: 8.95%.

*N*-(4-nonyloxybenxylidene)benzene-1,4-diamine, **2b**: *Yield* = 1.95 g (82.42%), mp: 135.6–137.7 °C, yellow powder. FTIR (cm^−1^): 3459 and 3330 (N-H stretching), 2921 and 2851 (C*sp^3^*-H stretching), 1606 (C=N stretching), 1509 (aromatic C=C stretching), 1248 (C-O stretching), 1170 (C-N stretching). ^1^H-NMR (500 MHz, DMSO-d_6_) δ, ppm: 8.47 (s, 1*H*), 7.79 (d, *J* = 10.0 Hz, 2H), 7.06 (d, *J* = 10.0 Hz, 2H), 7.00 (d, *J* = 5.0 Hz, 2H), 6.65 (d, *J* = 10.0 Hz, 2H), 6.43 (s, 2H), 4.05 (t, *J* = 7.50 Hz, 2H), 1.72–1.77 (m, 2H), 1.42–1.46 (m, 2H), 1.24–1.37 (m, 10H), 0.88 (t, *J* = 7.5 Hz, 3H). ^13^C-NMR (125 MHz, DMSO-d_6_) δ, ppm: 161.33, 154.65, 147.52, 141.40, 139.43, 130.03, 122.37, 115.37, 115.09, 68.57, 31.63, 29.28, 29.16, 29.12, 28.94, 25.90, 22.36, 14.07. CHN elemental analysis: Calculated for C_22_H_30_N_2_O: C: 78.06%, H: 8.93%, N: 8.28%; Found: C: 77.83%, H: 8.89%, N: 8.19%.

*N*-(4-decyloxybenxylidene)benzene-1,4-diamine, **2c**: *Yield* = 2.11 g (85.63%), mp: 134.3–136.1 °C, white powder. FTIR (cm^−1^): 3457 and 3336 (N-H stretching), 2919 and 2851 (C*sp^3^*-H stretching), 1603 (C=N stretching), 1509 (aromatic C=C stretching), 1251 (C-O stretching), 1170 (C-N stretching). ^1^H-NMR (500 MHz, DMSO-d_6_) δ, ppm: 8.47 (s, 1H), 7.79 (d, *J* = 5.0 Hz, 2H), 7.05 (d, *J* = 10.0 Hz, 2H), 7.00 (d, *J* = 5.0 Hz, 2H), 6.64 (d, *J* = 5.0 Hz, 2H), 6.42 (s, 2H), 4.05 (t, *J* = 7.50 Hz, 2H), 1.72–1.77 (m, 2H), 1.42–1.48 (m, 2H), 1.25–1.38 (m, 12H), 0.88 (t, *J* = 5.0 Hz, 3H). ^13^C-NMR (125 MHz, DMSO-d_6_) δ, ppm: 161.32, 154.61, 147.55, 141.37, 139.43, 130.02, 122.36, 115.38, 115.06, 68.58, 31.64, 29.31, 29.27, 29.16, 29.10, 28.98, 25.90, 22.35, 14.07. CHN elemental analysis: Calculated for C_23_H_32_N_2_O: C: 78.36%, H: 9.15%, N: 7.95%; Found: C: 78.13%, H: 9.19%, N: 7.99%.

*N*-(4-dodecyloxybenxylidene)benzene-1,4-diamine, **2d**: *Yield* = 2.15 g (80.83%), mp: 128.5–133.3 °C, yellow powder. FTIR (cm^−1^): 3462 and 3333 (N-H stretching), 2919 and 2851 (C*sp^3^*-H stretching), 1606 (C=N stretching), 1509 (aromatic C=C stretching), 1248 (C-O stretching), 1170 (C-N stretching). ^1^H-NMR (500 MHz, DMSO-d_6_) δ, ppm: 8.47 (s, 1H), 7.79 (d, *J* = 5.0 Hz, 2H), 7.05 (d, *J* = 10.0 Hz, 2H), 7.00 (d, *J* = 10.0 Hz, 2H), 6.64 (d, *J* = 5.0 Hz, 2H), 6.43 (s, 2H), 4.06 (t, *J* = 7.50 Hz, 2H), 1.72–1.78 (m, 2H), 1.42–1.48 (m, 2H), 1.25–1.39 (m, 16H), 0.88 (t, *J* = 7.5 Hz, 3H). ^13^C-NMR (125 MHz, DMSO-d_6_) δ, ppm: 161.32, 154.62, 147.55, 141.36, 139.26, 130.02, 122.37, 115.38, 115.06, 68.57, 31.64, 29.35, 29.33, 29.31, 29.29, 29.15, 29.08, 28.99, 25.88, 22.35, 14.07. CHN elemental analysis: Calculated for C_25_H_36_N_2_O: C: 78.90%, H: 9.53%, N: 7.36%; Found: C: 78.11%, H: 9.51%, N: 7.43%.

*N*-(4-tetradecyloxybenxylidene)benzene-1,4-diamine, **2e**: *Yield* = 2.32 g (81.23%), mp: 126.6–128.8 °C, yellow powder. FTIR (cm^−1^): 3459 and 3320 (N-H stretching), 2919 and 2849 (C*sp^3^*-H stretching), 1609 (C=N stretching), 1512 (aromatic C=C stretching), 1254 (C-O stretching), 1170 (C-N stretching). ^1^H-NMR (500 MHz, DMSO-d_6_) δ, ppm: 8.47 (s, 1H), 7.77 (d, *J* = 10.0 Hz, 2H), 7.06 (d, *J* = 10.0 Hz, 2H), 6.99 (d, *J* = 10.0 Hz, 2H), 6.58 (d, *J* = 10.0 Hz, 2H), 5.10 (s, 2H), 4.00 (t, *J* = 7.50 Hz, 2H), 1.68–1.73 (m, 2H), 1.37–1.42 (m, 2H), 1.18–1.33 (m, 20H), 0.83 (t, *J* = 7.5 Hz, 3H). ^13^C-NMR (125 MHz, DMSO-d_6_) δ, ppm: 161.32, 154.62, 147.55, 141.36, 139.24, 130.02, 122.36, 115.38, 115.06, 68.57, 31.64, 29.36, 29.35, 29.34, 29.33, 29.30, 29.29, 29.15, 29.08, 28.99, 25.88, 22.35, 14.07. CHN elemental analysis: Calculated for C_27_H_40_N_2_O: C: 79.36%, H: 9.87%, N: 6.86%; Found: C: 78.56%, H: 9.80%, N: 6.79%.

*N*-(4-hydroxybenxylidene)benzene-1,4-diamine, **2f**: *Yield* = 2.23 g (75.08%), mp: 186.2–188.4 °C, yellow powder. FTIR (cm^−1^): 3100–3350 and 3285 (N-H stretching), 3198 (O-H stretching), 1603 (C=N stretching), 1504 (aromatic C=C stretching), 1221 (C-O stretching), 1162 (C-N stretching). ^1^H-NMR (500 MHz, DMSO-d_6_) δ, ppm: 10.00 (s, 1H), 8.42 (s, 1H), 7.69 (d, *J* = 10.0 Hz, 2H), 7.04 (d, *J* = 5.0 Hz, 2H), 6.84 (d, *J* = 10.0 Hz, 2H), 6.58 (d, *J* = 10.0 Hz, 2H), 5.08 (s, 2H). ^13^C-NMR (125 MHz, DMSO-d_6_) δ, ppm: 160.55, 157.38, 156.12, 143.68, 130.63, 128.37, 122.59, 116.11, 116.03. CHN elemental analysis: Calculated for C_13_H_12_N_2_O: C: 73.56%, H: 5.70%, N: 13.20%; Found: C: 74.00%, H: 5.75%, N: 13.09%.

*N*-(4-carboxybenxylidene)benzene-1,4-diamine, **2g**: *Yield* = 2.57 g (76.49%), mp: 178.3–180.4 °C, yellow powder. FTIR (cm^−1^): 3150–3350 and 3309 (N-H stretching), 3210 (O-H stretching), 1698 (C=O stretching), 1593 (C=N stretching), 1536 (aromatic C=C stretching), 1385 (C-O stretching), 1149 (C-N stretching). ^1^H-NMR (500 MHz, DMSO-d_6_) δ, ppm: 8.67 (s, 1H), 8.02 (d, *J* = 5.0 Hz, 2H), 7.95 (d, *J* = 10.0 Hz, 2H), 7.19 (d, *J* = 10.0 Hz, 2H), 6.62 (d, *J* = 5.0 Hz, 2H). ^13^C-NMR (125 MHz, DMSO-d_6_) δ, ppm: 167.54, 153.36, 148.96, 141.05, 139.42, 130.12, 129.52, 128.32, 123.33, 114.61. CHN elemental analysis: Calculated for C_14_H_12_N_2_O_2_: C: 69.99%, H: 5.03%, N: 11.66%; Found: C: 69.36%, H: 5.06%, N: 11.52%.

*N*-(4-chlorobenxylidene)benzene-1,4-diamine, **2h**: *Yield* = 2.78 g (86.07%), mp: 196.7–198.3 °C, yellow powder. FTIR (cm^-1^): 3328 and 3217 (N-H stretching), 1617 (C=N stretching), 1504 (aromatic C=C stretching), 1168 (C-N stretching), 790 (C-Cl bending). ^1^H-NMR (500 MHz, DMSO-d_6_) δ, ppm: 8.59 (s, 1H), 7.87 (d, *J* = 5.0 Hz, 2H), 7.52 (d, *J* = 5.0 Hz, 2H), 7.15 (d, *J* = 5.0 Hz, 2H), 6.61 (d, *J* = 10.0 Hz, 2H), 5.26 (s, 2H). ^13^C-NMR (125 MHz, DMSO-d_6_) δ, ppm: 153.20, 148.68, 139.54, 136.20, 135.23, 129.94, 129.25, 123.11, 114.59. CHN elemental analysis: Calculated for C_13_H_11_N_2_: C: 67.68%, H: 4.81%, N: 12.14%; Found: C: 67.61%, H: 4.82%, N: 12.05%.

*N*-(4-nitrobenxylidene)benzene-1,4-diamine, **2i**: *Yield* = 4.34 g (85.75%), mp: 172.1–174.5 °C, red powder. FTIR (cm^−1^): 3484 and 3384 (N-H stretching), 1625 (C=N stretching), 1504 (aromatic C=C stretching), 1160 (C-N stretching). ^1^H-NMR (500 MHz, DMSO-d_6_) δ, ppm: 8.76 (s, 1H), 8.31 (d, *J* = 5.0 Hz, 2H), 8.09 (d, *J* = 5.0 Hz, 2H), 7.25 (d, *J* = 5.0 Hz, 2H), 6.63 (d, *J* = 5.0 Hz, 2H), 5.44 (s, 2H). ^13^C-NMR (125 MHz, DMSO-d_6_) δ, ppm: 151.66, 149.62, 148.40, 143.19, 138.85, 129.09, 124.43, 123.80, 114.50. CHN elemental analysis: Calculated for C_13_H_11_N_3_O_2_: C: 64.72%, H: 4.60%, N: 17.42%; Found: C: 64.62%, H: 4.62%, N: 17.21%.

#### 3.3.3. Synthesis of Hexakis(4-Formlyphenoxy)cyclotriphosphazene, **3**

4-Hydroxybenzaldehyde (9.77 g, 0.08 mol) and triethylamine (10.12 g, 0.1 mol) were dissolved in 150 mL acetone and the mixture was cooled to 0 °C in an ice bath. The mixture was stirred for 30 min. Then, a solution of hexachlorocyclotriphosphazene (3.48 g, 0.01 mol) in 50 mL acetone was added dropwise to the mixture. A white salt began to precipitate within few minutes. After 2 h at 0 °C, the reaction was allowed to attain at room temperature and was continued to stir for an additional 94 h. The reaction was monitored using TLC. Upon completion, the mixture was poured into 250 mL of cold water and left overnight in the fridge. The precipitate formed was then filtered, washed with water, and dried. The precipitate was recrystallised from methanol. Yield: 8.11 g (94.19%), mp: 200.8–204.2 °C, white powder. FTIR (cm^−1^): 2730 (H-C=O), 1699 (C=O stretching), 1595 (C=C stretching), 1205 (C-O stretching), 1151 (P=N stretching), 944 (P–O–C stretching). ^1^H-NMR (500 MHz, DMSO-d_6_) δ, ppm: 9.90 (s, 1H), 7.78 (d, *J* = 10.0 Hz, 2H), 7.16 (d, *J* = 10.0 Hz, 2H). ^13^C-NMR (125 MHz, DMSO-d_6_) δ, ppm: 191.69, 153.59, 133.55, 131.46, 121.03. ^31^P-NMR (500 MHz, DMSO-d_6_) δ, ppm: 7.60 (s, 1P). CHN elemental analysis: Calculated for C_42_H_30_N_3_O_12_P_3_: C: 58.55%, H: 3.51%, N: 4.88%; Found: C: 58.28%, H: 3.48%, N: 4.83%.

#### 3.3.4. Synthesis of Hexakis{4-((*E*)-((4-(((*E*)-4-substituted-benzylidene)amino)phenyl)imino)methyl) phenoxy}triazaphosphazene, **4a**–**i**

Hexakis{4-((*E*)-((4-(((*E*)-4-heptyloxy-benzylidene)amino)phenyl)imino)methyl)phenoxy} triazaphosphazene, **4a**: Intermediate **3** (0.50 g, 0.58 mmol) and intermediate **2a** (1.26 g, 4.07 mmol) were mixed in 25 mL methanol in a 100 mL round bottom flask. A few drops of glacial acetic acid was added to the mixture which was stirred and refluxed for 24 h. The reaction progress was monitored by TLC. Upon completion, the mixture was cooled in an ice bath. The precipitate formed was filtered, washed with cold methanol, and dried. The crude was recrystallised from ethanol. The same method was used to synthesize **4b**–**i**. Yield: 1.12 g (73.68%), mp: 130.5–133.1 °C, yellow powder. FTIR (cm^−1^): 2924 and 2857 (C*sp^3^*-H stretching), 1613 (C=N stretching), 1509 (C=C stretching), 1248 (C-O stretching), 1192 (P=N stretching), 1168 (C-N stretching), 983 (P–O–C stretching). ^1^H-NMR (500 MHz, DMSO-d_6_) δ, ppm: 8.66 (s, 1H) 8.55 (s, 1H), 7.95 (d, *J* = 10.0 Hz, 2H), 7.87 (d, *J* = 10.0 Hz, 2H), 7.55 (d, *J* = 5.0 Hz, 2H), 7.33 (d, *J* = 10.0 Hz, 2H), 7.28 (d, *J* = 15.0 Hz, 2H), 7.04 (d, *J* = 5.0 Hz, 2H), 4.07 (t, *J* = 5.0 Hz, 2H), 1.73–1.79 (m, 2H), 1.42–1.48 (m, 2H), 1.27–1.38 (m, 6H), 0.90 (t, *J* = 7.5 Hz, 3H). ^13^C-NMR (125 MHz, DMSO-d_6_) δ, ppm: 162.09, 162.02, 159.38, 158.83, 150.63, 149.14, 136.49, 135.76, 130.81, 130.53, 129.28, 122.32, 122.13, 115.46, 68.64, 31.58, 29.16, 28.76, 25.88, 22.31, 14.06. ^31^P-NMR (500 MHz, DMSO-d_6_) δ, ppm: 8.20 (s, 1P). CHN elemental analysis: Calculated for C_162_H_174_N_15_O_12_P_3_: C: 74.37%, H: 6.70%, N: 8.03%; Found: C: 74.29%, H: 6.68%, N: 7.95%.

Hexakis{4-((*E*)-((4-(((*E*)-4-nonyloxy-benzylidene)amino)phenyl)imino)methyl)phenoxy} triazaphosphazene, **4b**: Yield: 1.13 g (70.19%), mp: 128.1–130.5 °C, yellow powder. FTIR (cm^−1^): 2919 and 2851 (C*sp^3^*-H stretching), 1616 (C=N stretching), 1509 (C=C stretching), 1254 (C-O stretching), 1195 (P=N stretching), 1168 (C-N stretching), 981 (P–O–C stretching). ^1^H-NMR (500 MHz, DMSO-d_6_) δ, ppm: 8.66 (s, 1H) 8.55 (s, 1H), 7.95 (d, *J* = 10.0 Hz, 2H), 7.87 (d, *J* = 5.0 Hz, 2H), 7.55 (d, *J* = 10.0 Hz, 2H), 7.33 (d, *J* = 10.0 Hz, 2H), 7.28 (d, *J* = 15.0 Hz, 2H), 7.04 (d, *J* = 5.0 Hz, 2H), 4.07 (t, *J* = 5.0 Hz, 2H), 1.73–1.78 (m, 2H), 1.42–1.48 (m, 2H), 1.26–1.38 (m, 10H), 0.88 (t, *J* = 5.0 Hz, 3H). ^13^C-NMR (125 MHz, DMSO-d_6_) δ, ppm: 162.09, 162.02, 159.06, 158.48, 150.63, 149.14, 136.48, 135.77, 130.81, 130.54, 129.29, 122.32, 122.09, 115.46, 68.63, 31.64, 29.29, 29.14, 28.94, 25.90, 22.36, 14.08. ^31^P-NMR (500 MHz, DMSO-d_6_) δ, ppm: 8.18 (s, 1P). CHN elemental analysis: Calculated for C_174_H_198_N_15_O_12_P_3_: C: 75.05%, H: 7.17%, N: 7.55%; Found: C: 74.88%, H: 7.10%, N: 7.48%.

Hexakis{4-((*E*)-((4-(((*E*)-4-decyloxy-benzylidene)amino)phenyl)imino)methyl)phenoxy} triazaphosphazene, **4c**: *Yield* = 1.15 g (69.28%), mp: 125.7–127.4 °C, yellow powder. FTIR (cm^−1^): 2919 and 2851 (C*sp^3^*-H stretching), 1613 (C=N stretching), 1509 (C=C stretching), 1251 (C-O stretching), 1195 (P=N stretching), 1168 (C-N stretching), 983 (P–O–C stretching). ^1^H-NMR (500 MHz, DMSO-d_6_) δ, ppm: 8.66 (s, 1H) 8.55 (s, 1H), 7.96 (d, *J* = 5.0 Hz, 2H), 7.87 (d, *J* = 5.0 Hz, 2H), 7.55 (d, *J* = 10.0 Hz, 2H), 7.33 (d, *J* = 10.0 Hz, 2H), 7.28 (d, *J* = 15.0 Hz, 2H), 7.04 (d, *J* = 10.0 Hz, 2H), 4.07 (t, *J* = 7.5 Hz, 2H), 1.73–1.79 (m, 2H), 1.43–1.48 (m, 2H), 1.26–1.38 (m, 12H), 0.88 (t, *J* = 7.5 Hz, 3H). ^13^C-NMR (125 MHz, DMSO-d_6_) δ, ppm: 162.09, 162.02, 159.06, 158.48, 150.63, 149.15, 136.48, 135.76, 130.75, 130.54, 129.29, 122.33, 122.09, 115.46, 68.63, 31.64, 29.29, 29.15, 29.13, 28.94, 25.90, 22.36, 14.08. ^31^P-NMR (500 MHz, DMSO-d_6_) δ, ppm: 8.20 (s, 1P). CHN elemental analysis: Calculated for C_180_H_210_N_15_O_12_P_3_: C: 75.37%, H: 7.38%, N: 7.32%; Found: C: 75.12%, H: 7.31%, N: 7.27%.

Hexakis{4-((*E*)-((4-(((*E*)-4-dodecyloxy-benzylidene)amino)phenyl)imino)methyl)phenoxy} triazaphosphazene, **4d**: *Yield* = 1.23 g (69.89%), mp: 121.2–123.7 °C, yellow powder. FTIR (cm^−1^): 2919 and 2849 (C*sp^3^*-H stretching), 1616 (C=N stretching), 1509 (C=C stretching), 1248 (C-O stretching), 1192 (P=N stretching), 1168 (C-N stretching), 980 (P–O–C stretching). ^1^H-NMR (500 MHz, DMSO-d_6_) δ, ppm: 8.67 (s, 1H) 8.56 (s, 1H), 7.96 (d, *J* = 5.0 Hz, 2H), 7.87 (d, *J* = 10.0 Hz, 2H), 7.56 (d, *J* = 10.0 Hz, 2H), 7.34 (d, *J* = 15.0 Hz, 2H), 7.28 (d, *J* = 10.0 Hz, 2H), 7.05 (d, *J* = 5.0 Hz, 2H), 4.08 (t, *J* = 7.5 Hz, 2H), 1.73–1.79 (m, 2H), 1.43–1.49 (m, 2H), 1.26–1.38 (m, 16H), 0.88 (t, *J* = 5.0 Hz, 3H). ^13^C-NMR (125 MHz, DMSO-d_6_) δ, ppm: 162.09, 162.03, 159.48, 158.56, 149.82, 149.15, 136.57, 135.76, 130.83, 130.56, 129.31, 122.34, 122.15, 115.49, 68.64, 31.64, 29.31, 29.26, 29.13, 29.11, 29.09, 28.97, 25.88, 22.35, 14.08. ^31^P-NMR (500 MHz, DMSO-d_6_) δ, ppm: 8.19 (s, 1P). CHN elemental analysis: Calculated for C_192_H_234_N_15_O_12_P_3_: C: 75.93%, H: 7.77%, N: 6.92%; Found: C: 75.77%, H: 7.73%, N: 6.88%.

Hexakis{4-((*E*)-((4-(((*E*)-4-tetradecyloxy-benzylidene)amino)phenyl)imino)methyl)phenoxy} triazaphosphazene, **4e**: *Yield* = 1.36 g (74.73%), mp: 118.8–120.5 °C, yellow powder. FTIR (cm^−1^): 2916 and 2849 (C*sp^3^*-H stretching), 1616 (C=N stretching), 1512 (C=C stretching), 1251 (C-O stretching), 1195 (P=N stretching), 1170 (C-N stretching), 973 (P–O–C stretching). ^1^H-NMR (500 MHz, DMSO-d_6_) δ, ppm: 8.66 (s, 1H) 8.54 (s, 1H), 7.95 (d, *J* = 10.0 Hz, 2H), 7.87 (d, *J* = 10.0 Hz, 2H), 7.54 (d, *J* = 10.0 Hz, 2H), 7.33 (d, *J* = 5.0 Hz, 2H), 7.27 (d, *J* = 10.0 Hz, 2H), 7.03 (d, *J* = 10.0 Hz, 2H), 4.08 (t, *J* = 7.5 Hz, 2H), 1.73–1.79 (m, 2H), 1.43–1.49 (m, 2H), 1.23–1.40 (m, 16H), 0.88 (t, *J* = 10.0 Hz, 3H). ^13^C-NMR (125 MHz, DMSO-d_6_) δ, ppm: 162.18, 162.14, 159.33, 158.38, 150.70, 159.04, 136.54, 135.86, 130.74, 130.54, 129.28, 122.31, 122.07, 115.56, 68.77, 31.68, 29.38, 29.34, 29.20, 29.18, 29.14, 29.12, 29.02, 29.00, 25.92, 22.34, 14.01. ^31^P-NMR (500 MHz, DMSO-d_6_) δ, ppm: 8.21 (s, 1P). CHN elemental analysis: Calculated for C_204_H_258_N_15_O_12_P_3_: C: 76.44%, H: 8.11%, N: 6.55%; Found: C: 76.32%, H: 8.07%, N: 6.49%.

Hexakis{4-((*E*)-((4-(((*E*)-4-hydroxy-benzylidene)amino)phenyl)imino)methyl)phenoxy} triazaphosphazene, **4f**: *Yield* = 1.21 g (73.33%), mp: 212.5–215.5 °C, light brown powder. FTIR (cm^−1^): 3221 (OH stretching), 1617 (C=N stretching), 1493 (C=C stretching), 1264 (C-O stretching), 1189 (P=N stretching), 1160 (C-N stretching), 981 (P–O–C stretching). ^1^H-NMR (500 MHz, DMSO-d_6_) δ, ppm: 10.11 (s, 1H), 8.71 (s, 1H) 8.51 (s, 1H), 7.97 (d, *J* = 10.0 Hz, 2H), 7.78 (d, *J* = 10.0 Hz, 2H), 7.61 (d, *J* = 5.0 Hz, 2H), 7.37 (d, *J* = 15.0 Hz, 2H), 7.28 (d, *J* = 20.0 Hz, 2H), 6.88 (d, *J* = 10.0 Hz, 2H). ^13^C-NMR (125 MHz, DMSO-d_6_) δ, ppm: 156.64, 156.50, 155.77, 155.64, 142.33, 142.19, 135.42, 135.32, 129.90, 129.76, 128.79, 122.72, 122.59, 115.71. ^31^P-NMR (500 MHz, DMSO-d_6_) δ, ppm: 8.24 (s, 1P). CHN elemental analysis: Calculated for C_120_H_90_N_15_O_12_P_3_: C: 71.10%, H: 4.48%, N: 10.37%; Found: C: 70.76%, H: 4.44%, N: 10.28%.

Hexakis{4-((*E*)-((4-(((*E*)-4-carboxy-benzylidene)amino)phenyl)imino)methyl)phenoxy} triazaphosphazene, **4g**: *Yield* = 1.23 g (69.10%), mp: 195.2–197.1 °C, yellow powder. FTIR (cm^−1^): 3219 (OH stretching), 1673 (C=O stretching), 1619 (C=N stretching), 977 (C=C stretching), 1291 (C-O stretching), 1195 (P=N stretching), 1173 (C-N stretching), 1012 (P–O–C stretching). ^1^H-NMR (500 MHz, DMSO-d_6_) δ, ppm: 10.14 (s, 1H), 8.69 (s, 1H) 8.51 (s, 1H), 7.90 (d, *J* = 5.0 Hz, 2H), 7.78 (d, *J* = 10.0 Hz, 2H), 7.74 (d, *J* = 10.0 Hz, 2H), 7.37 (d, *J* = 15.0 Hz, 2H), 7.28 (d, *J* = 20.0 Hz, 2H), 6.88 (d, *J* = 10.0 Hz, 2H). ^13^C-NMR (125 MHz, DMSO-d_6_) δ, ppm: 164.41, 164.40, 157.70, 155.10, 148.79, 142.55, 142.11, 140.04, 129.51, 126.59, 124.36, 123.57, 116.31, 116.22. ^31^P-NMR (500 MHz, DMSO-d_6_) δ, ppm: 8.23 (s, 1P). CHN elemental analysis: Calculated for C_126_H_90_N_15_O_18_P_3_: C: 68.94%, H: 4.13%, N: 9.57%; Found: C: 68.77%, H: 4.09%, N: 9.48%.

Hexakis{4-((E)-((4-(((E)-4-chloro-benzylidene)amino)phenyl)imino)methyl)phenoxy} triazaphosphazene, **4h**: *Yield* = 1.33 g (76.44%), mp: 201.1–204.4 °C, yellow powder. FTIR (cm^−1^): 1618 (C=N stretching), 1515 (C=C stretching), 1273 (C-O stretching), 1192 (P=N stretching), 1165 (C-N stretching), 983 (P–O–C stretching), 823 (C-Cl bending). ^1^H-NMR (500 MHz, DMSO-d_6_) δ, ppm: 8.80 (s, 1H) 8.64 (s, 1H), 8.30 (d, *J* = 10.0 Hz, 2H), 8.17 (d, *J* = 10.0 Hz, 2H), 7.93 (d, *J* = 10.0 Hz, 2H), 7.53 (d, *J* = 10.0 Hz, 2H), 7.41 (d, *J* = 15.0 Hz, 2H), 7.34 (d, *J* = 15.0 Hz, 2H). ^13^C-NMR (125 MHz, DMSO-d_6_) δ, ppm: 159.28, 158.94, 158.53, 158.17, 142.22, 142.15, 136.70, 135.52, 130.64, 129.98, 129.32, 124.22, 122.67, 122.42. ^31^P-NMR (500 MHz, DMSO-d_6_) δ, ppm: 8.25 (s, 1P). CHN elemental analysis: Calculated for C_120_H_84_Cl_6_N_15_O_6_P_3_: C: 67.42%, H: 3.96%, N: 9.83%; Found: C: 67.33%, H: 3.92%, N: 9.78%.

Hexakis{4-((*E*)-((4-(((*E*)-4-nitro-benzylidene)amino)phenyl)imino)methyl)phenoxy} triazaphosphazene, **4i**: *Yield* = 3.06 g (79.90%), mp: 183.7–186.7 °C, orange powder. FTIR (cm^−1^): 1618 (C=N stretching), 1520 (C=C stretching), 1273 (C-O stretching), 1195 (P=N stretching), 1165 (C-N stretching), 981 (P–O–C stretching). ^1^H-NMR (500 MHz, DMSO-d_6_) δ, ppm: 8.85 (s, 1H) 8.68 (s, 1H), 8.34 (d, *J* = 5.0 Hz, 2H), 8.21 (d, *J* = 5.0 Hz, 2H), 7.97 (d, *J* = 5.0 Hz, 2H), 7.57 (d, *J* = 5.0 Hz, 2H), 7.44 (d, *J* = 15.0 Hz, 2H), 7.37 (d, *J* = 15.0 Hz, 2H). ^13^C-NMR (125 MHz, DMSO-d_6_) δ, ppm: 166.30, 157.70, 155.10, 150.40, 148.78, 142.55, 142.09, 136.98, 131.11, 129.51, 124.36, 124.09, 123.57, 116.30. ^31^P-NMR (500 MHz, DMSO-d_6_) δ, ppm: 8.20 (s, 1P). CHN elemental analysis: Calculated for C_120_H_84_N_21_O_18_P_3_: C: 65.48%, H: 3.85%, N: 13.36%; Found: C: 65.37%, H: 3.88%, N: 13.29%.

#### 3.3.5. Synthesis of Hexakis{4-((*E*)-((4-(((*E*)-4-amino-benzylidene)amino)phenyl)imino)methyl)phenoxy} triazaphosphazene, **4j**

Compound **4i** (2.50 g, 1.14 mmol) was dissolved in 20 mL methanol and a solution of sodium sulphide hydrate, Na_2_S.9H_2_0 (0.89 g, 0.01 mol) in 20 mL ethanol was mixed in 100 mL round bottom flask. The mixture was stirred and refluxed for 12 h. The reaction was monitored using TLC. Upon completion, the mixture was cooled in an ice bath and the precipitate formed was filtered, washed with cold ethanol, and dried overnight to give a red precipitate. Yield: 1.14 g (49.57%), mp: 198.2–200.8 °C, red powder. FTIR (cm^−1^): 3478 and 3312 (N-H stretching), 1617 (C=N stretching), 1493 (C=C stretching), 1273 (C-O stretching), 1186 (P=N stretching), 1162 (C-N stretching), 1002 (P–O–C stretching). ^1^H-NMR (500 MHz, DMSO-d_6_) δ, ppm: 8.67 (s, 1H) 8.50 (s, 1H), 7.89 (d, *J* = 10.0 Hz, 2H), 7.78 (d, *J* = 10.0 Hz, 2H), 7.72 (d, *J* = 10.0 Hz, 2H), 7.34 (d, *J* = 15.0 Hz, 2H), 7.27 (d, *J* = 15.0 Hz, 2H), 6.90 (d, *J* = 10.0 Hz, 2H). ^13^C-NMR (125 MHz, DMSO-d_6_) δ, ppm: 160.55, 158.83, 158.68, 149.43, 149.27, 135.46, 131.81, 130.43, 130.34, 127.89, 124.84, 121.94, 121.57, 115.70. ^31^P-NMR (500 MHz, DMSO-d_6_) δ, ppm: 8.22 (s, 1P). CHN elemental analysis: Calculated for C_120_H_96_N_21_O_6_P_3_: C: 71.31%, H: 4.79%, N: 14.55%; Found: C: 70.99%, H: 4.77%, N: 14.42%.

## 4. Conclusions

Intermediates **1a**–**e**, **2a**–**i**, and **3**, and final compounds **4a**–**j**, with two Schiff base linking units were successfully synthesized and characterized. All the intermediates and compound **4j** were found to be non-mesogenic, whereas other final compounds **4a**–**i** were mesogenic molecules. Compound **4a** showed smectic A and nematic phases while compounds **4b**–**d** exhibited only smectic A in heating and cooling cycles. In addition, compound **4e** exhibited smectic C and smectic A phases in both cycles. Meanwhile, all the other compounds with a small substituent such as hydroxy, carboxy, chloro, and nitro groups at the terminal end were found to exhibit a nematic phase. Further study on the fire-retardant properties of the hexasubstituted cyclotriphosphazene compounds was done using LOI testing. Interestingly, all the final compounds showed LOI values higher compared to that of polyester resin incorporated with HCCP. The highest LOI value obtained was 28.37%, which belonged to compound **4i** with the nitro terminal group. The presence of the NO_2_ group as an electron-withdrawing side arm of the cyclotriphosphazene compound reduces the exothermicity of the materials, which enhanced the fire-retardant property of the polyester resin.

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
