# Peer review of "Liquid-Crystal and Fire-Retardant Properties of New Hexasubstituted Cyclotriphosphazene Compounds with Two Schiff Base Linking Units"

_molecules, 2020, doi:10.3390/molecules25092122_

Round 1

Reviewer 1 Report

The manuscript “Liquid crystal and fire retardant properties of new hexasubstituted cyclotriphosphazene compounds with two Schiff base linking units” by Zuhair Jamain, Melati Khairuddean and Tay Guan-Seng is a large research. This is especially true for the synthetic part of the work.

All substances were fully characterized by IR, NMR, CHN, POM, and DSC methods (unfortunately the authors did not indicate whether crystals were obtained for intermediate compounds 2a-j because structural data for these compounds are not available in the Cambridge structural database).

The manuscript left a good impression and deserves to be published after some a moderate correction.

It should explaned for what reason the strategy of synthesis of target products through the formation of Schiff bases was chosen. It is also necessary to disclose in more detail why substituted HCCPs are excellent models for structure-activity studies.

In the section "Results and Discussion" it would be logical to start the discussion with the synthesis methodology and, hence, place all the preparative schemes for intermediate and final products at the beginning of this large section. This will make it easier for readers to perceive the description of the physicochemical study presented in the article. For the same purposes, it is better, to give a small conclusion in each subsection.

It would be nice to describe the liquid crystal behavior of the target materials in the form of a comparison with the best representatives in this field.

There are several comments on the experimental part. The methodology for LOI determination is not very clearly described, especially the calculation. The authors have to either expand the description or provide a link to an article where it is described in detail.

In the case of the calorimetric experiments, it is necessary to bring all the obtained curves into the supporting information.

A few words about English (typos and style).

The abstract requires careful rewriting. It should not look like part of an experimental section.

I would like the authors to use synonyms more often, rather than constantly repeating the same verbs and nouns (confirmed, showed and compounds).

Line 15-16: Hexasubstituted cyclotriphosphazene compounds, 4a-j, consist of two Schiff base linking units and different terminal substituents ….

Line 34: HCCP? (derivatives?) are excellent models for

Line 48: in his study [15]

Lines 63-71: put all verbs in past tens

89: need a comma before “which”

93: base linking units, 4a-i. The According

95: The stretching corresponding

327-329: BETTER to explain like rhis:The longer the alkyl tails are attached to the "inorganic" ring, the combustible are the target materials.

352: LOI = CF + k.d - correct multiplication sign

Scheme 3: change Y by X.

386: how many times extraction was performed?

Author Response

Response to Reviewer 1 Comments

Point 1: All substances were fully characterized by IR, NMR, CHN, POM, and DSC methods (unfortunately the authors did not indicate whether crystals were obtained for intermediate compounds 2a-j because structural data for these compounds are not available in the Cambridge structural database).

Response 1: For compounds 2a-i, the compounds did not show any liquid crystal phase. The statement already mentioned in line 241-242.

Line 241-242: In this study, all the intermediates (1a-e, 2a-j and 3) were found to be non-mesogonic with no liquid crystal behaviour.

Point 2: It should explaned for what reason the strategy of synthesis of target products through the formation of Schiff bases was chosen. It is also necessary to disclose in more detail why substituted HCCPs are excellent models for structure-activity studies.

Response 2: The reason and strategy for the target products and more details on HCCPs already mentioned in line 55-65 and line 67-76.

Line 55-65: HCCP molecule is a well-known compound with good thermal stability and fire retardancy, which due to the hexa-functionality and high phosphorus content. Modification of HCCP core system with organic side arms allow the exploration of new discotic molecules in liquid crystal field having fire retardant properties. Incorporation of HCCP into organic compound with Schiff base linking unit is to increase the resistance of the material towards ignition. The different terminal substituents in the side arms will gain a better insight of the structure-properties relationship of these types of compounds. The research study on liquid crystal materials with fire retardant properties having flexible ordered structures has led to the discovery of new properties and applications such as advanced technological devices or composites applications. Extensive exploration of these materials is able to detect small changes in temperature, electromagnetic radiation, mechanical stress and chemical environment.

Line 67-76:Schiff base contains carbon double bond with nitrogen atom connected to an alkyl or aryl group but not the hydrogen. These Schiff base molecules provide a stepped core structure which can maintain the molecular linearity. This linearity will provide high stability and enabling mesophase formation [25]. Besides, Schiff base linking units was found to enhance the properties of fire-retardant of polyester resin due to their thermal stability [26]. During the burning process, this molecule able to transform into a cross-linked structure, which promoted the formation of char on the surface in the condensed phase. These char layer shielded the surface of polyester resin from combustion [27]. Hence, both liquid crystal and fire retardant properties of cyclotriphosphazene core system attached with Schiff base linking unit were investigated in this study.

Point 3: In the section "Results and Discussion" it would be logical to start the discussion with the synthesis methodology and, hence, place all the preparative schemes for intermediate and final products at the beginning of this large section. This will make it easier for readers to perceive the description of the physicochemical study presented in the article. For the same purposes, it is better, to give a small conclusion in each subsection.

Response 3: The preparative schemes in section 3.3 already moved to section 2.1. Synthesis of the intermediates and final compounds in line 84. A small conclusion in each subsection was also added.

Point 4: It would be nice to describe the liquid crystal behavior of the target materials in the form of a comparison with the best representatives in this field.

Response 4: The comparison of the liquid crystal was added in Section 2.6 from line 312-364. The table for POM observation of compounds 4a-j was added for easy comparison. 

Point 5: There are several comments on the experimental part. The methodology for LOI determination is not very clearly described, especially the calculation. The authors have to either expand the description or provide a link to an article where it is described in detail.

Response 5: The description for LOI testing already added in section 3.2. in line 412-417.

Line 412-417: About 1 wt% of methyl ethyl ketone peroxide (MEKP) curing agent was added to the mixture and stirred until the sample is homogeneous and then poured into the moulds. The samples were cured for 5 hours in an oven at 60°C and left overnight at room temperature before it was burned using LOI testing. The LOI test was performed using an FTT oxygen index, according to BS 2782: Part 1: Method 141 and ISO 4589 with the dimension of 120 mm x 10 mm x 4 mm.

Point 6: In the case of the calorimetric experiments, it is necessary to bring all the obtained curves into the supporting information.

Response 6: All the DSC thermograms of compounds 4a-i are mentioned in the supplementary materials section.

Point 7: The abstract requires careful rewriting. It should not look like part of an experimental section.

Response 7: The abstract already revised in line 14-28.

Point 8: I would like the authors to use synonyms more often, rather than constantly repeating the same verbs and nouns (confirmed, showed and compounds).

Response 8: Some of the synonym verbs and nouns such as indicated, displayed, appeared, molecules and derivatives.

Point 9: Line 15-16: Hexasubstituted cyclotriphosphazene compounds, 4a-j, consist of two Schiff base linking units and different terminal substituents ….

Response 9: Line 14-15: A series of new hexasubstituted cyclotriphosphazene compounds (4a-j) consist of two Schiff base linking units and different terminal substituents ...

Point 10: Line 34: HCCP? (derivatives?) are excellent models for ...

Response 10: Line 34: HCCP derivatives are excellent models for ...

Point 11: Line 48: ... in his study [15].

Response 11: Line 47-48: ... fire retardants [16].

Point 12: Lines 63-71: put all verbs in past tens

Response 12: All the verbs already changed to past tense 

Line 77-82: In this work, polyester resin (PE) was used as a matrix for moulding to study the fire retardant properties of these compounds. PE is an unsaturated synthetic resin and can be considered as a combustible material [28]. This resin produces a lot of heat during combustion due to low thermal stability and becomes a potential fire risk. Thus, the modification of polyester resin by mixing with cyclotriphosphazene-based compounds having fire retardant properties is needed to overcome this problem.

Point 13: Line 89: need a comma before “which”

Response 13: Line 128: ... intermediates, 2a-i, which confirmed ...

Point 14: Line 93: base linking units, 4a-i. The According

Response 14: Line 135: According to ...

Point 15: Line 95: The stretching corresponding

Response 15: Line 143: The stretching corresponding ...

Point 16: Line 327-329: BETTER to explain like rhis:The longer the alkyl tails are attached to the "inorganic" ring, the combustible are the target materials.

Response 16: Line 391-392: The longer the alkyl tails are attached to the cyclotriphosphazene ring, the combustible are the target materials [53, 54].

Point 17: Line 352: LOI = CF + k.d - correct multiplication sign

Response 17: Line 420: LOI = CF + (k x d)

Point 18: Scheme 3: change Y by X.

Response 18: Already changed to X as shown in Scheme 2 (Line 88)

Point 19: Line 386: how many times extraction was performed?

Response 19: There are 3 times extraction was performed as shown in line 433: ...was extracted using dichloromethane (3x30 mL).

Reviewer 2 Report

This is a carefully conducted study on the liquid crystal and fire retardant properties of a number of hexakis substituted cyclotriphosphazenes. The presentation is clear and logic. Conclusions appear justified. However, the autors should pay attention to several points before final acceptance is possible. Two of my comments are major ones.

Firstly,

Line 74:  I think that it is necessary to begin the discussion of results by somehow describing the compounds used in this study. This could be made by referring to Schemes 1-4 already in the beginning of presentation of results, although the detailed presentation of synthesis takes place later in chapter 3.3. Readers must see the structures that the formula numbers refer to. The syntheses as such are rather conventional text book reactions, and that is why this overall presentation of the synthesis strategy can be rather short.

Secondly

Line 535: The name of the final compounds must be wrong. It refers only to two aromatic moieties: benzylidene and phenyl. However, in each substituent, there are three benzene rings. My suggestion is:

hexakis{4-((E)-((4-(((E)-4-substituted benzylidene)amino)phenyl)imino)methyl)phenoxy}triazaphosphazene. The names of individual compounds 4a-j should be corrected accordingly.

Then several minor points:

Line 17:   4-carboxyphenyl- instead of benzoic acid

Line 32: Hexachlorocyclotriphosphazene

Lines 50-53: The sentence should be reformulated. In case I have understood correctly, the purpose is to say:  The chemical reactions that take place in the condensed phase at elevated temperature, such as hydrolysis, dehydration and chain scission, produce carbonaceous char residues on the surface of the phosphorus-based materials which retards the material from further burning.

Line 65:  ….polyester resin (PE) is used….

Line 66:  PE is an unsaturated….

Line 69: ….is needed…

Lines 70-71:  I don´t understand the purpose

Line 89: ….in any of the intermediates 2a-i….

Line 93: The According …

Lines 101-102: 4-carboxyphenyl instead of benzoic acid

Line 128: …..consisting

Line 136: absent instead of disappeared

Line 169: are instead of were

Line 247: 4-carboxyphenyl instead of benzoic acid

Lines 258-260: …..all final compounds in the nine series with a different linking unit that bore a terminal alkoxy function were mesogenic and the width of the thermal mesomorphic range was increased with increasing chain length.

Lines 269-270: Small terminal substituents, such as hydroxyl or carboxy, do not……

Line 274: ..was

Line 359: …alkyl bromides…

Scheme 3: A1 rather than A1

Scheme 4: A11 rather than A11

Line 381: 4-Heptyloxybenzaldehyde (capital letter after a number in the beginning of sentence)

Line 387: …with anhydrous….

Line 519: hexakis(4-formylphenoxy)cyclotriphosphazene

Author Response

Response to Reviewer 2 Comments

Point 1: Line 74:  I think that it is necessary to begin the discussion of results by somehow describing the compounds used in this study. This could be made by referring to Schemes 1-4 already in the beginning of presentation of results, although the detailed presentation of synthesis takes place later in chapter 3.3. Readers must see the structures that the formula numbers refer to. The syntheses as such are rather conventional text book reactions, and that is why this overall presentation of the synthesis strategy can be rather short.

Response 1: All the schemes in section 3.3 already moved in new subtopic in section 2.1. Synthesis of the intermediates and final compounds in line 84-110. Scheme 2 and 3 are combined into one Scheme only.

Point 2: Line 535: The name of the final compounds must be wrong. It refers only to two aromatic moieties: benzylidene and phenyl. However, in each substituent, there are three benzene rings. My suggestion is:

hexakis{4-((E)-((4-(((E)-4-substituted benzylidene)amino)phenyl)imino)methyl)phenoxy}triazaphosphazene. The names of individual compounds 4a-j should be corrected accordingly.

Response 2: All the compounds name already changed as suggested by reviewer as shown in line 584-587, 603-604, 616-617, 629-630, 642-643, 655-656, 666-667, 676-677, 687-688 and 698-699.

Point 3: Line 17:   4-carboxyphenyl- instead of benzoic acid

Response 3: Line 19: ... hydroxy, 4-carboxyphenyl, ...

Point 4: Line 32: Hexachlorocyclotriphosphazene

Response 4: Line 32: Hexachlorocyclotriphosphazene

Point 5: Lines 50-53: The sentence should be reformulated. In case I have understood correctly, the purpose is to say:  The chemical reactions that take place in the condensed phase at elevated temperature, such as hydrolysis, dehydration and chain scission, produce carbonaceous char residues on the surface of the phosphorus-based materials which retards the material from further burning.

Response 5: Line 50-52: The chemical reactions that take place in the condensed phase at elevated temperature, such as hydrolysis, dehydration and chain scission, produce carbonaceous char residues on the surface of the phosphorus-based materials which retards the material from further burning [18,19].

Point 6: Line 65:  ….polyester resin (PE) is used….

Response 6: Line 77: In this work, polyester resin (PE) was used as....

Before correction: Line 65: In this research, polyester resin (PE) will be used as a ...

Point 7: Line 66:  PE is an unsaturated….

Response 7: Line 78: PE is an unsaturated ...

Point 8: Line 69: ….is needed…

Response 8: Line 81: ... fire retardant properties is needed to overcome ...

Point 9: Lines 70-71:  I don´t understand the purpose

Response 9: The statement already removed from the text.

Point 10: Line 89: ….in any of the intermediates 2a-i….

Response 10: Line 128: ...observed in any of the intermediates, 2a-i,...

Point 11: Line 93: The According …

Response 11: Line 135: According to ...

Point 12: Lines 101-102: 4-carboxyphenyl instead of benzoic acid

Response 12: Line 140: ...of the 4-carboxyphenyl substituent...

Point 13: Line 128: …..consisting

Response 13: Line 172: These signals consist of ...

Point 14: Line 136: absent instead of disappeared

Response 14: Line 183: ... which were absent in...

Point 15: Line 169: are instead of were

Response 15: Line 214: ... spectra are summarised in Table 1.

Point 16: Line 247: 4-carboxyphenyl instead of benzoic acid

Response 16: Line 300: ... 4-carboxyphenyl, chloro ...

Point 17: Lines 258-260: …..all final compounds in the nine series with a different linking unit that bore a terminal alkoxy function were mesogenic and the width of the thermal mesomorphic range was increased with increasing chain length.

Response 17: Line 312-314: all the final compounds in the series with a Schiff base linking unit that bore a different terminal alkoxy function were mesogenic and the wider of the thermal mesomorphic range was increased with increasing chain length.

Point 18: Lines 269-270: Small terminal substituents, such as hydroxyl or carboxy, do not……

Response 18: Line 332: Small terminal substituents, such as hydroxy or carboxy, do not...

Point 19: Line 274: ..was

Response 19: Line 337: ... terminal end also exhibited...

Point 20: Line 359: …alkyl bromides…

Response 20: Line 86: ...different alkyl bromides...

Point 21: Scheme 3: A1 rather than A1

Response 21:  Line 98: The symbol already changed to A1

Point 22: Scheme 4: A11 rather than A11

Response 22: Line 102: The symbol already changed to A11

Point 23: Line 381: 4-Heptyloxybenzaldehyde (capital letter after a number in the beginning of sentence)

Response 23: Line 428: 4-Heptyloxybenzaldehyde, 1a: 4-Hydroxybenzaldehyde (all the corrections have been made in section 3.3)

Point 24: Line 387: …with anhydrous…

Response 24: Line 434: ...dried with anhydrous...

Point 25: Line 519: hexakis(4-formylphenoxy)cyclotriphosphazene

Response 25: Line 568: Synthesis of hexakis(4-formlyphenoxy)cyclotriphosphazene, 3

Round 2

Reviewer 2 Report

The authors have adequately revised the presentation according to the suggestions made in the first reviewers report. I recommend acceptance in the present form.I